# First Report, Characterization and Pathogenicity of *Vibrio chagasii* Isolated from Diseased Reared Larvae of Chilean Scallop, *Argopecten purpuratus* (Lamarck, 1819)

**DOI:** 10.3390/pathogens12020183

**Published:** 2023-01-24

**Authors:** Rocío Urtubia, Claudio D. Miranda, Sergio Rodríguez, Javier Dubert, Juan L. Barja, Rodrigo Rojas

**Affiliations:** 1Programa Doctorado en Acuicultura, Facultad de Ciencias del Mar, Universidad Católica del Norte, Coquimbo 1780000, Chile; 2Laboratorio de Patobiología Acuática, Departamento de Acuicultura, Universidad Católica del Norte, Coquimbo 1780000, Chile; 3Departamento de Microbiología y Parasitología, CIBUS—Instituto de Acuicultura, Universidad de Santiago de Compostela, 15782 Santiago de Compostela, Spain; 4Centro AquaPacífico, Coquimbo 1780000, Chile

**Keywords:** Vibriosis, *Vibrio chagasii*, scallop larvae, mollusk pathogen, aquaculture, Chile

## Abstract

Two *Vibrio* strains (VPAP36 and VPAP40) were isolated from moribund-settled larvae of the Chilean scallop *Argopecten purpuratus* during vibriosis outbreaks that occurred in two commercial scallop larvae hatcheries located in the Inglesa and Tongoy bays in Northern Chile. The strains were identified as *Vibrio chagasii* using phenotypic characterization and whole genome sequence analysis. Both strains exhibited the phenotypic properties associated with virulence, gelatin hydrolysis and β-hemolysis, whereas only VPAP36 produced phospholipase and only VPAP40 produced caseinase. The whole genome analysis showed that the strains harbored genes encoding for the virulence factors, the EPS type II secretion system, and Quorum Sensing (auto-inductor 1 and auto-inductor 2), whereas genes encoding a metalloproteinase and a capsular polysaccharide were detected only in the VPAP40 genome. When challenge bioassays using healthy 11-day-old scallop larvae were performed, the *V. chagasii* VPAP36 and VPAP40 strains exhibited significant (*p* < 0.05) differences in their larval lethal activity, producing, after 48 h, larval mortalities of 65.51 ± 4.40% and 28.56 ± 5.35%, respectively. Otherwise, the cell-free extracellular products of the VPAP36 and VPAP40 strains produced larval mortalities of 20.86 ± 2.40% and 18.37 ± 2.40%, respectively, after 48 h of exposure. This study reports for the first time the isolation of *V. chagasii* from the massive larval mortalities of the farmed scallop (*Argopecten purpuratus*) in Chile, and demonstrates the pathogenic activity of *V. chagasii* towards the Chilean scallop, the second most important species for Chilean mariculture.

## 1. Introduction

The high rate of growth of the aquaculture industry has caused an emerging spread of associated bacterial diseases to been increasingly observed, characterized by a predominance of vibriosis outbreaks [1,2,3,4]. Vibriosis is commonly defined as any sort of disease produced by the *Vibrio* species in the infected host [5,6].

The culture of the Chilean scallop *Argopecten purpuratus* (Lamarck, 1819) is one of the most commercially important industries of Chilean mariculture and is mainly concentrated in the northern region of the country [7]. Although scallop larval production have been successfully developed in Chilean hatcheries, episodes of massive mortalities of reared-larvae, mostly caused by bacterial pathogens, occasionally occur [8,9,10,11]. This high prevalence of bacterial infections consists mainly of Vibriosis episodes, which significantly reduce spat production and increase operational costs, prompting the need for an extensive use of antimicrobials, mainly florfenicol and oxytetracycline [12,13,14].

Several studies have demonstrated important occurrences of scallop larval mortalities produced by the pathogenic activity of various *Vibrio* species in Chilean commercial hatcheries [9,10,11,15,16]. Previous reports have shown the incidence of *Vibrio anguillarum*-related and *V. alginolyticus*-exhibiting pathogenic activity among reared scallop larvae [15,16]. More recently, the isolation of the *V. splendidus* and *V. bivalvicida* species from moribund-reared scallop larvae from commercial hatcheries of the Chilean scallop has been reported, and their pathogenic activity on this shellfish species has been demonstrated [9,10]. Furthermore, in a more recent study, a representative of the *V. europaeus* species was reported for the first time in the Pacific Ocean, and it exhibited a high virulence towards Chilean scallop larvae [11].

Strains belonging to the *Vibrio chagasii* species have been isolated from various sources, including seawater, marine sediment, sea bass, turbot larvae, rotifers, and Artemia [17,18,19]. More recently, the *V. chagasii* species was identified as a causal agent of disease in scallops, oysters, and mussels [20,21,22], but it has never been reported as an agent causing mollusk larval mortalities.

In the present research, we identified for the first time two representatives of the *V. chagasii* species associated with the massive mortality of scallop larvae (*Argopecten purpuratus*); thus, the main aim of the study was to describe their phenotypic and genomic characteristics, as well as to demonstrate their pathogenic activity towards scallop larvae under controlled conditions.

The study of pathogenic *Vibrio* strains causing massive larval mortalities may provide relevant information about the most frequently encountered *Vibrio* species and associated vibriosis events with the Chilean pectinid industry. This knowledge will be useful in establishing efficient sanitary management programs in mollusk hatcheries, as was previously discussed [14,23,24].

## 2. Materials and Methods

### 2.1. Bacterial Isolation

The bacterial strains VPAP36 and VPAP40 were isolated from settled moribund larval samples of the Chilean scallop (*Argopecten purpuratus*) during two different mass mortality events that occurred in two geographically distant commercial hatcheries located in the north of Chile, in Inglesa Bay (27°05′30″ S; 70°51′51″ W) and Tongoy Bay (30°15′22″ S; 71°29′46″ W), respectively (Figure 1). Triplicate larval samples were aseptically collected from the bottom of the culture ponds during water exchange using a sterile container, transported to the Aquatic Pathobiology Lab of the Universidad Católica del Norte, and immediately processed. The larval samples were centrifuged at 2000× *g* using an Eppendorf Model 5415D centrifuge and homogenized in 5 mL of sterile seawater (0.2 μm). The homogenized sample obtained was inoculated in triplicate on Soy Trypticase agar supplemented with 2% NaCl (TSA2, Becton-Dickinson, Sparks, MD, USA), and on Thiosulfate-Citrate-Bile-Salts Sucrose agar (TCBS, Becton-Dickinson) prepared with 50% aged micro-filtered (0.45 μm) seawater. The plates were incubated at 20 °C for 48 h. The predominant colonies that developed were purified using TSA2 and stored at −85 °C in CryoBank^TM^ vials (Mast Diagnostica, Reinfeld, Germany). The strains were grown on TSA2 agar at 20 °C for 24 h prior to use.

### 2.2. Phenotypic Characterization

The VPAP36 and VPAP40 strains were phenotypically characterized according to the method suggested by Noguerola and Blanch for *Vibrio* species [25]. The phenotypic tests, Gram staining, cell morphology, oxidation/fermentation (O/F) of glucose, production of oxidase and catalase, and susceptibility to the vibriostatic agent O/129 (2,4-diamino-6,7-diisopropylpteridine) (10 and 150 μg per disc) (Bio-Rad, Marnes-la-Coquette, France) were determined according to the procedures described in Buller [26], in media supplemented with NaCl (2%). Furthermore, several key characteristics for the description of bacterial strains belonging to the *Vibrionaceae* family, such as their growth on TCBS and MacConkey (Becton-Dickinson) agar, Møller’s L-lysine and L-ornithine decarboxylase, and Thornley’s arginine dihydrolase, their hydrolysis of aesculin, their indole production, and their growth at different temperatures and concentrations of NaCl, were assessed. Growth at different temperatures (4, 20, 30, 35, 40 °C) was tested on Tryptic soy broth (Difco labs.) supplemented with 2% NaCl, whereas growth at different salinities (0%, 3%, 6%, 8%, and 10% of NaCl) was tested on peptone broth (Difco), using procedures as previously described [27,28].

Furthermore, the VPAP36 and VPAP40 strains were phenotypically characterized using the API 20E (bioMérieux, Marcy-l’Etoile, France), and strains were additionally screened for their ability to utilize the 95 organic substrates inoculating the GN2 microplates of the Biolog system (Biolog Inc., Hayward, CA, USA) assay. Both systems were inoculated according to the manufacturers’ instructions with some modifications. For the API 20E system, the *Vibrio* strains were resuspended in saline (0.85% NaCl), inoculated, and incubated at 20 °C for 48 h, as suggested by MacDonnell et al. [29]. For the Biolog system, the strains were resuspended in a solution containing 2.5% NaCl, 0.8% MgCl_2_, and 0.05% KCl and incubated at 20 °C for 72 h. For the Biolog system, strains were inoculated using a solution containing 2.5% NaCl, 0.8% MgCl_2_ and 0.05% KCl according to the instructions of the manufacturer, the microplates were incubated aerobically in the dark at 20 °C, and duplicate readings were made after 48 h and 72 h of incubation, but only the results after 72 h of incubation were considered. Additional phenotypic characteristics were determined using the API 20E system (bioMérieux, Marcy-l’Etoile, France).

Additional enzymatic activities of the VPAP36 and VPAP40 strains were determined using the API ZYM system (bioMérieux, Marcy-l’Etoile, France) according to the manufacturer’s guidelines. Briefly, isolates were cultured overnight in Soy Trypticase Broth supplemented with 2% NaCl (TSB2, Becton-Dickinson) for 24 h at 20 °C and centrifuged at 3300× *g* at 4° C, after which the pellet obtained was resuspended in a sterile 0.9% (w/v) NaCl solution to obtain a turbidity of 5-6 McFarland (1.5–1.8 × 10^9^ mL^−1^), and 65 µL of each suspension were added to each cupule. The test strips were incubated for 4 h at 20 °C. The test strips were read after 5 min as indicated by the manufacturer, and each assay was performed twice to ensure reproducibility. The results were recorded according to the following classification: 0, negative reaction; 1–2, weak activity; 3–5, strong activity.

### 2.3. Virulence Factors Production

The production of the virulence factors caseinase, gelatinase, lipase, β-hemolysin, and phospholipase was determined as described by Natrah et al. [30]. For the lipase and phospholipase assays, marine agar 2216 (MA, Becton-Dickinson) plates supplemented with 1% Tween 80 (Sigma-Aldrich, Darmstadt, Germany) and 1% egg yolk emulsion (Oxoid, Hampshire, United Kingdom), respectively, were used. The caseinase and gelatinase assays were performed using plates with double-strength MA supplemented with a 4% skim milk powder suspension (Oxoid) and plates with MA with 0.5% gelatin (Sigma-Aldrich) added, respectively. The β-hemolysis of sheep erythrocytes was determined using Columbia Blood agar (Oxoid). All assays were performed in triplicate.

### 2.4. Bacterial DNA Extraction and Sequencing

For DNA extraction, bacterial isolates were cultured in Marine Agar for 24 h and resuspended in PBS. The DNA extractions were performed from a pure culture using the DNeasy Blood and Tissue kit (QIAGEN, Venlo, The Netherlands), following the manufacturer’s instructions. The obtained DNA extractions were quantified using a Qubit 3 Fluorometer (Thermo Fisher Scientific, Waltham, MA, USA), and their quality was ensured by the measurement of the A260/A280 and A260/230 ratios using a NanoDrop One Spectrophotometer (Thermo Scientific). Five μL of the extractions were run on a 2% agarose gel to ensure the integrity of the genomes. The library preparation for the genomic sequencing was based on the Nanopore protocol, Rapid sequencing gDNA-barcoding (SQK-RBK004). The sequencing of the genomes was performed using the Oxford Nanopore MinION.

### 2.5. Bacterial Identification

The obtained sequences were assembled using the PATRIC server, through Unicycler [31]. Also, similar genomes were searched using Mash/MinHash [32]. The whole genomic DNA sequences were used to identify the strains through a genome comparison using the PATRIC bioinformatics platform. The VPAP36 and VPAP40 strains were identified by a comparison analysis of 100 single-copy genes using the PATRIC server (https://patricbrc.org/app/PhylogeneticTree, Accessed on 21 September 2022), which analyzes the aligned proteins and coding DNA from single-copy genes using the program RAxML [33]. The genomic comparison of the VPAP36 and VPAP40 strains with the 21 related members of the Splendidus, Orientalis, and Harveyi clades currently available was performed. OrthoAni analysis [34] was performed to quantify the overall sequence similarity. The draft genomes of the two selected strains have been submitted to the NCBI database under accession numbers JAOZWY000000000 (VPAP36) and JAPDFM000000000 (VPAP40).

### 2.6. *Genomic Analysis of Virulence Factors*

The *Vibrio chagasii* VPAP36 and VPAP40 genomes were analyzed to detect genes associated with virulence factors using the PATRIC platform [35,36].

### 2.7. Pathogenic Activity

The pathogenic activity of the VPAP36 and VPAP40 strains was determined by performing a challenge with healthy larvae of *A. purpuratus*. Larvae of 130.5 ± 15.6 μm (11 day-old) were added to each well of a microplate of 12 wells (Orange Scientifique, Braine-l’Alleud, Belgium) with 4 mL of micro-filtered seawater (0.2 μm) to obtain a final concentration of 20–25 Larvae mL^−1^. The larvae were exposed in triplicate to a concentration of 2.1 × 10^5^ CFU mL^−1^ of the VPAP36 and VPAP40 pathogenic strains and of the non-pathogenic strain *Vibrio* sp. VNPAP02, included as a control, in the challenge assay. In addition, a group of non-inoculated larvae was included in the assay. The microplates were incubated at 18 °C for 48 h in darkness. The percentage of live and dead larvae in each of the treatments was determined every 12 h, and the clinical signs manifested by the larvae were recorded using a Nikon Model Eclipse Ts2 inverted microscope. The larvae were considered dead when they showed no ciliar activity or leaflet movement.

### 2.8. Pathogenic Activity of Extracellular Products of Vibrio chagasii

The extracellular products produced by the pathogenic strains VPAP36 and VPAP40 were obtained by the plate cellophane technique [37] by disseminating 0.1 mL of a logarithmic culture grown in Soy Trypticase Broth supplemented with 2% NaCl (TSB2) on sterile cellophane sheets located on the surface of the culture media. The plaques were incubated for 36–48 h, and the bacterial cells grown on the cellophane were washed with PBS. The obtained suspensions were centrifuged at 13,000 rpm for 30 min at 4 °C, and the supernatants were filtered (0.2 μm) and stored at −20 °C. The concentration of proteins in the supernatants was determined using the Micro-BCA (Pierce) assay according to the manufacturer’s instructions. The scallop larvae (120–130 μm) kept in microplates of 12 wells at a density of 20 larvae mL^−1^ were inoculated in triplicate with the supernatant of the *Vibrio* strains to obtain a concentration of 4 μg of total mL^−1^ proteins and were observed at 12, 24, 36, and 48 h under an inverted microscope.

### 2.9. Statistical Analyses

For larval assays, the mortality percentages were transformed to arcsin (square root [mortality rate ration]) and were analyzed using one-way analysis of variance (ANOVA). The normality of the variables was determined using the Kolmogorov–Smirnov test, whereas the homogeneity of the variances was determined using Levene’s test [38]. When the overall differences were significant (*p* < 0.05), a Tukey’s multiple range test was used to determine significant differences (*p* < 0.05) among the proportions of the mortality of the challenged and control scallop larvae. All statistical analyses were performed using the Sigma version 3.1 computer program (Systat Software Inc., San José, CA, USA).

## 3. Results

### 3.1. Phenotypic *Characterization*

The bacterial strains VPAP36 and VPAP40 showed the phenotypic characteristics typical of members of the *Vibrio* species [39]. The studied strains were Gram-negative, short, motile rod-shaped oxidase and catalase producers susceptible to O/129 (10 and 150 µg), showed positive growth on TCBS agar, and required NaCl for growth. The growth of the strains occurred at 20–35 °C, but not at 4 and 40 °C, and they cannot tolerate 8 and 10% NaCl. Both strains were positive for arginine dihydrolase and negative for lysine decarboxylase and ornithine decarboxylase.

When some enzymatic properties associated with bacterial virulence were determined, the VPAP36 strain produced phospholipase and gelatinase and possessed β-hemolytic activity, while the VPAP40 strain produced caseinase and gelatinase and showed β-hemolytic activity, and neither of strains were lipase producers. The API ZYM profiles of the VPAP36 and VPAP40 strains are presented in Table 1**,** showing the capacity of both isolates to produce the enzymes alkaline phosphatase, leucine arylamidase, valine arylamidase, trypsin, acid phosphatase, and naphthol-AS-BI-phosphohydrolase, whereas only the VPAP40 strain was able to produce β-Glucoronidase.

When the API 20E system was used, both strains were positive for arginine dihydrolase, citrate utilization, indole production, gelatinase, and the fermentation of glucose, mannitol, saccharose, and amygdalin, whereas only the VPAP36 strain produced tryptophane deaminase. Further phenotypic characterization of *Vibrio* strains indicated a high phenotypical homogeneity showing the utilization of the following BIOLOG substrates as sole carbon sources: dextrin, glycogen, N-acetyl-D-galactosamine, D-cellobiose, D-fructose, maltose, D-mannitol, sucrose, D-trehalose, D,L-lactic acid, succinic acid, L-alanine, L-alanyl-glycine, L-asparagine, L-aspartic acid, L-glutamic acid, glycil-L-aspartic acid, glycyl-L-glutamic acid, L-proline, L-serine, L-threonine, inosine, uridine, thymidine, glycerol, glucose-1-phosphate, and glucose-6-phosphate. Otherwise, only the VPAP36 strain utilized α-D-glucose, D-mannose, acetic acid, D-galacturonic acid, α-keto glutaric acid, quinic acid, bromo succinic acid, and L-histidine, whereas only the VPAP40 strain utilized tween 40, tween 80, α-keto valeric acid, D-alanine, and D,L-α-glycerol phosphate. The phenotypic characteristics of the VPAP36 and VPAP40 strains obtained using the API 20E and Biolog systems are listed in Appendix A, respectively.

### 3.2. Virulence Factors Production

Both *V. chagasii* strains produced gelatinase and β-hemolysin but were negative for lipase production. Phospholipase was only produced by the VPAP36 strain, whereas caseinase was only produced by the VPAP40 strain.

### 3.3. Bacterial Identification

The whole genome sequences of the VPAP36 and VPAP40 strains were compared with several whole genome sequences currently available for related members of the *Splendidus*, *Orientalis* and *Harveyi* clades. The results are presented as a phylogenetic dendrogram, as is depicted in Figure 2, showing that both Chilean strains are members of the genus *Vibrio*, being genetically most closely related to the *Vibrio chagasii* strains.

When the six publicly available genomes of *V. chagasii* deposited in GenBank (LMG21353, CCUG 48643, M14-00480, M14-00614, M14-00606, M14-00606b) were analyzed with the OrthoANI software, used for calculating average nucleotide identity, the genomes of the Chilean strains were most closely related to the genomes of the *V. chagasii* M14-00606 and M14-00614 strains, isolated from summer mortalities in Pacific oysters at Port Stephens, NSW, Australia. The VPAP36 strain showed a similarity of 98.03% with the strain *V. chagasii* M14-00614, whereas the VPAP40 strain exhibited a similarity of 97.92% with the strain *V. chagasii* M14-00606, as shown in Figure 3, thus confirming the identity of both Chilean strains as belonging to the *Vibrio chagasii* species.

### 3.4. *Genomic Analysis of Virulence Factors*

Through the analysis of bacterial genomes, it was possible to detect several genes associated with virulence in the genomes of both *V. chagasii* strains, including the EPS type II secretion system, which is a membrane-bound protein complex used to secrete a variety of different proteins, such as toxins and degradative enzymes and genes associated with the quorum sensing system (auto-inductor 1 and auto-inductor 2). Other genes detected only in the VPAP40 genome include genes encoding a metalloproteinase and a capsular polysaccharide, whereas a gene encoding a thermo-labile hemolysin was observed in the VPAP36 genome.

### 3.5. *Pathogenic Activity*

The VPAP36 and VPAP40 strains showed pathogenic activity on the challenged healthy scallop larvae, producing the classical signs of vibriosis affecting mollusk larvae. These signs were identical to those observed in the larval culture suffering the vibriosis outbreaks developed in the commercial hatcheries when the VPAP36 and VPAP40 strains were recovered. The main clinical symptoms exhibited by larvae infected with the *V. chagasii* strains were distension of the velum, and accumulation of bacterial swarms on the margins of the larvae, which appeared at 24 h post-infection (Figure 4).

The pathogenic activity of the strains was demonstrated by infecting healthy scallop larvae with both pathogenic *V. chagasii* strains and the non-pathogenic *Vibrio* sp. VNPAP02 strain, demonstrating that the VPAP36 and VPAP40 strains produced high and moderate levels of larval mortality, respectively, after 48 h post-infection (Figure 5A). The VPAP36 strain produced significantly (*p* < 0.05) higher levels of larval mortality than those produced by the VPAP40 strain after a period of 12-48 h of infection (Figure 5A). After 24 and 48 h of exposure, the larval mortality of the larvae challenged with the VPAP36 strain was 38.94 ± 3.77% and 65.51 ± 4.40%, respectively, which was significantly (*p* < 0.05) higher than that observed in the larvae challenged with the VPAP40 (2.471 ± 1.94% and 28.56 ± 5.35%, respectively). Not challenged control larvae exhibited a mortality of 2.5 ± 1.00% after a period of 48 h, which was not significantly different than that produced by the non-pathogenic VNPAP02 strain (5.80 ± 2.10%).

When scallop larvae were exposed to extracellular products (ECPs) produced by the VPAP36 and VPAP40 strains, they exhibited identical symptoms to those observed during the bacterial challenges, mainly characterized by the distension of the velum and the accumulation of bacterial aggregates on the margins of the larvae, as shown in Figure 4. The mortality rates of the larvae infected with ECPs produced by the VPAP36 and VPAP40 strains remained at levels not significantly different (*p* < 0.05), and, after 12 h of exposure, no dead larvae were observed (Figure 5B). At 24 h post-inoculation with the ECPs of the VPAP36 and VPAP40 strains, the percentages of larval mortality were 4.29 ± 0.70% and 1.92 ± 0.63%, respectively. At 48 h post-inoculation with ECPs of the VPAP36 and VPAP40 strains, the percentages of larval mortality were 20.86 ± 2.40 and 18.37 ± 2.4%, respectively, which were significantly different (*p* < 0.05) from the larval mortality of the control group (1.5 ± 0.5%) (Figure 5B).

## 4. Discussion

Bacterial diseases, mainly vibriosis outbreaks, have been reported to be the main causes of mass mortality in the larval culture of bivalve mollusks [40,41]. In Chile, infectious bacterial outbreaks have caused high mortalities in the larval culture of *A*. *purpuratus*, resulting in high economic losses. Several species of *Vibrio* have been reported to be the causative agents of these infectious outbreaks [9,10,11,15,16], but information related to pathogenic bacteria in the culture of *A. purpuratus* is still limited.

The scallop is one of the most economically important mollusks in Chilean aquaculture, and seed production is based on both natural capture and supply from the controlled culture in hatcheries, but natural capture depends on environmental conditions, being restricted to certain months of the year, and does not ensure the acquisition of the required quantity and quality seeds. Unfortunately, the supply of hatchery seeds is limited by the frequent occurrence of bacterial infections that cause mass mortalities that significantly reduce production and increase operational costs [14].

In this study, two strains identified as representatives of the *V*. *chagasii* species were isolated from the mortalities of A. *purpuratus* larvae, and their pathogenic activity was demonstrated in a scallop larvae challenge assay. This is the first report of *V. chagasii* in Chilean mariculture and the first worldwide report of the pathogenic role of this *Vibrio* species in shellfish larvae. According to the latest update to the phylogenetic analyses of bacteria belonging to the Vibrionaceae family, *V. chagasii* corresponds to the *Splendidus* clade [42]. *V. chagasii* was proposed as a new species by Thompson et al. [17], along with the species *V. kanaloae* and *V. pomeroyi*. These strains are phylogenetically related to *V. splendidus*, but DNA-DNA hybridization experiments proved that they belong to three novel species [43,44,45]. *V. chagasii* has been isolated from seawater samples [19] and intestine samples of turbot (*Scophthalmus maximus*) larvae [17], as well as from natural biofilms [21] and rotifer cultures [46]. Furthermore, *V. chagasii* has recently been identified as a causal agent of disease in scallops, oysters, and mussels [20,21,22], but has never been reported to causing mortality in the larval stages of mollusks.

Previously, *V*. *chagasii* was reported as a causative agent of adult pectinid *Patinopecten yessoensis* mortality, the main clinical sign of which was the presence of abscesses in the adductor muscle [20]. Furthermore, a *V. chagasii* strain isolated from natural biofilm carried 23 genes related to virulence, including genes encoding for a hemolysin, OmpW, and the gene *fur*, associated with iron assimilation, and the authors suggested that the strain has a pathogenic potential capable of causing infections in adult mussels [21]. In another strain belonging to the *V. chagasii* species, isolated from gilthead seabream larvae cultured in Portugal, various genes associated with virulence, disease, and defense were detected, including genes involved in cholera toxin biosynthesis, toxic compounds, and active host invasion and intracellular resistance [4]. In accordance with these results, the Chilean *V. chagasii* strains carried many genes associated with virulence, including the virulence factors metalloproteinase, hemolysins, and the EPS type II secretion system, among others. Considering that Vibrionaceae have been associated with various pathological processes in mollusk culture, and that this group presents a high taxonomic diversity, one of the pending aspects to be investigated corresponds to the characterization of the genes associated with virulence factors such as the expression of metalloproteases, hemolysins, siderophores, and others. Other genes detected in the analyzed genomes include the gene *fur*, encoding a histidine-rich ferric-uptake regulator (Fur) protein, previously detected in *V. cholerae* and *V. vulnificus* [47,48], as well as extensively detected in the Vibrionaceae family [49]. The ability of pathogenic bacteria to capture iron in the host organism contributes greatly to their virulence capacity, since it facilitates their growth inside the host [50]; thus, genes related to iron uptake such as *fur* regulate the expression of virulence genes [51].

Symptoms produced by infection with *V. chagasii* strains have also been observed as results of the infection of larvae with strains of *V. bivalvicida* and *V. europaeus* species [10,11], but several symptoms observed after 48 h of infection by these strains, such as erratic larval swimming, disruption of the velum, the detachment of ciliary cells from the velum, and necrosis of the digestive gland tissue, were not observed in larvae infected with the *V. chagasii* species. This would be due to the higher virulence of the *V. bivalvicida* and *V. europaeus* strains, which caused mortality percentages of 97.4 ± 1.2% and 90.03 ± 6.8%, respectively, well above the larval mortality percentages caused by the VPAP36 and VPAP40 strains (65.5 ± 4.40 and 28.6 ± 5.4%, respectively).

When the levels of larval mortality produced by the pathogenic *V. chagasii* strains in the challenge assays using *A. purpuratus* larvae are compared with those produced by other pathogenic *Vibrio* species isolated from outbreaks that occurred in Chilean scallop hatcheries, both *V. chagasii* strains exhibit significantly lower percentages of larval mortality than those produced by the *V. bivalvicida* and *V. europaeus* strains, and they are mainly characterized by the distension of the velum and the accumulation of bacterial aggregates on the margins of the larvae [10,11]. The ECPs produced by the *V. chagasii* strains were less toxic to larvae than whole bacterial cells, which is in agreement with was previously reported for virulent strains belonging to the *V. bivalvicida*, *V. europaeus*, and *V. tubiashii* species, suggesting that various virulence factors are required to produce a high degree of larval mortality [10,11,52]. The most likely explanation for this difference is that it is due to pathogenic activity mediated by tissue degradation and the invasiveness of the living cells of *V. chagasii*. Previous studies using challenged blue-mussel (*Mytilus edulis*) and Manila clam (*Ruditapes philippinarum*) larvae infected by distinct GFP-tagged *Vibrio* species demonstrated the importance of a progressive invasive activity during the infectious disease [53,54]. It was observed that the *Vibrio* cells were filtered by larvae, entering in the digestive system through the esophagus and colonizing the digestive gland, and rapidly expanding to the surrounding organs, completely colonizing the larvae [54]. Apparently, virulence factors such as the production of a phospholipase by the VPAP36 strain would favor the development of the invasive process within the larva. It is well known that microbial phospholipases are involved in several pathogenic processes, either acting directly, by causing severe cell damage and the consequent destruction of membranes, or by helping the bacteria to evade the larval immune defense mechanisms, thus favoring the pathogen’s invasive activity. Furthermore, several soluble virulence factors, including caseinase and hemolysins, indicate the ability of these strains to induce larval tissue damage, favoring bacterial cell invasion and the progression of the disease.

In another study, Rojas et al. [9] isolated three strains of *V. splendidus* from massive mortalities of *A. purpuratus* larvae that occurred in Chilean commercial hatcheries and confirmed their pathogenic activity, which was mainly mediated by non-protein, thermostable extracellular components, observing important differences in their abilities to produce larval mortality. The authors concluded that, to avoid culture collapse, concentrations of the *V. splendidus* species of below 10^4^ CFU per mL should be maintained in the larval-rearing tanks [9]. However, this guideline would be highly dependent on the virulence of the vibrio strain present in the larval culture. These *V. splendidus* strains produced lower levels of larval mortality after 48 h of exposure than those produced by the *V. bivalvicida* and *V. europaeus* strains [9,10,11], but they were similar to the mortality percentages produced by the VPAP36 strain, which were significantly higher than those produced by the VPAP40 strain.

It is evident that more studies are required to achieve the isolation of the infectious agents that cause recurrent episodes of mass mortality in the larval and juvenile phases of farmed scallops in Chile and to verify their pathogenic activity through the development of bacterial challenge bioassays under controlled conditions. As previously reported [14,23,24,55], the implementation of efficient health management programs in shellfish larvae hatcheries is of great importance in reducing mortality in the culture. Thus, there is a prevailing need to implement larval quality indicators in mollusk mass culture systems that allow the premature detection of culture deterioration that favors the development of opportunistic bacterial pathologies, especially those caused by virulent vibrios. Under this premise, Miranda et al. [56] developed a manual that describes the procedures for the measurement of sanitary, physiological, and nutritional indicators in larval cultures of *A. purpuratus*.

## 5. Conclusions

This study reports for the first time the isolation in Chilean pectiniculture of two pathogenic strains belonging to the *V. chagasii* species, recovered from two massive larval mortality episodes that occurred in two distant commercial scallop hatcheries. In addition, we demonstrated for the first time in experimentally challenged scallop larvae a pathogenic effect of the *V. chagasii* strains against *A. purpuratus*. It was observed that the *V. chagasii* strains produced similar symptoms to those observed during the massive mortality events that occurred in the Chilean commercial scallop hatcheries, confirming the role of the *V. chagasii* strains as the causative agents of the observed disease outbreaks in reared scallop larvae. Our results demonstrated that *V. chagasii* are virulent to scallop larvae, exhibiting moderate (VPAP36 strain) and low (VPAP40 strain) virulence. The difference in the levels of virulence between the viable cells of both *V. chagasii* strains is probably due to the efficacy of their invasive activity as well as the production of extracellular products, which could enhance the bacterial invasion process. It is concluded that the *V. chagasii* species is an emergent pathogen in Chilean pectinid hatcheries, because of their demonstrated capacity to cause mass mortalities of reared scallop larvae, thus making them a pathogen of major concern for this industry. Thus, their identification in larval cultures is essential for developing efficient sanitary control measures to prevent larval losses in Chilean intensive scallop larvae husbandry.

## Figures and Tables

**Figure 1 pathogens-12-00183-f001:**
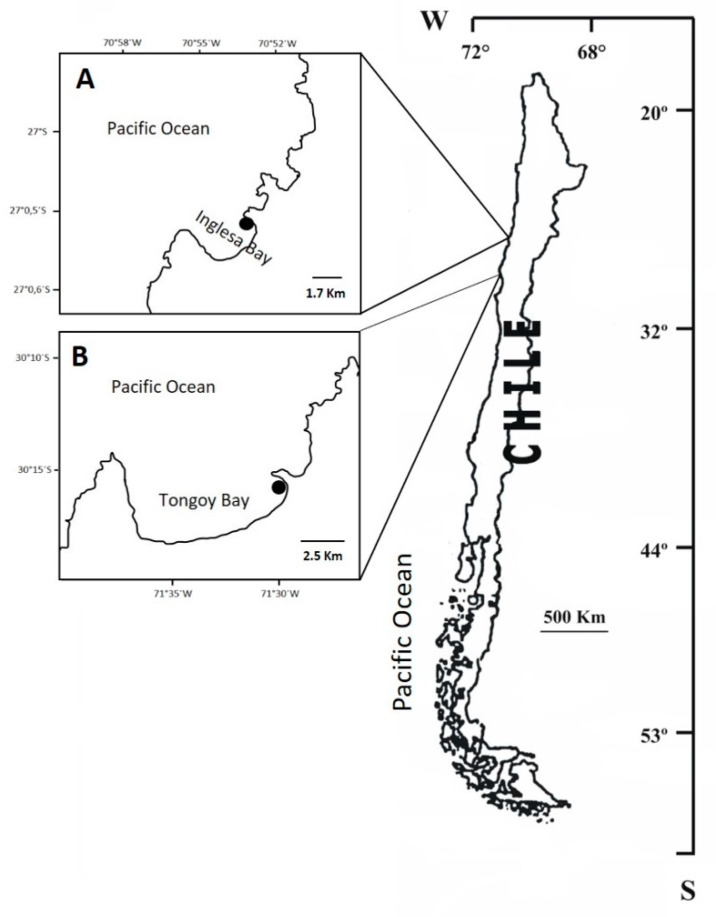
The geographic location of the scallop hatcheries where the *C. chagasii* strains VPAP36 (**A**) and VPAP40 (**B**) were recovered.

**Figure 2 pathogens-12-00183-f002:**
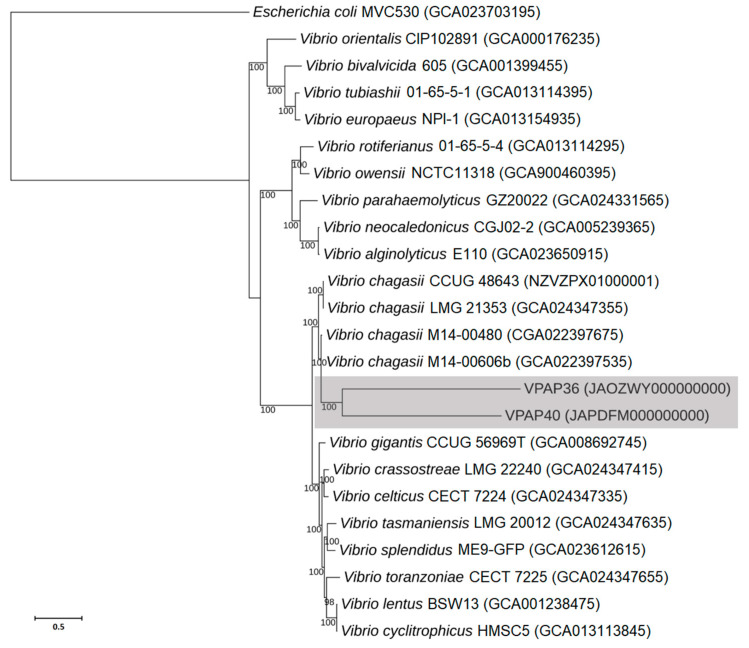
Phylogenetic tree constructed by the Maximum Likelihood method based on the whole genome sequences, showing the relationship between the Chilean *Vibrio chagasii* strains VPAP36 and VPAP40 and related taxa within the Splendidus, Orientalis, and Harveyi clades. *Escherichia coli* MVC530 (GenBank Assembly Accession: GCA_023703195) was used as an outgroup. Accession numbers of sequences are shown in parentheses. Horizontal branch lengths are proportional to evolutionary divergences. Bootstrap values (%) appear next to the corresponding branch.

**Figure 3 pathogens-12-00183-f003:**
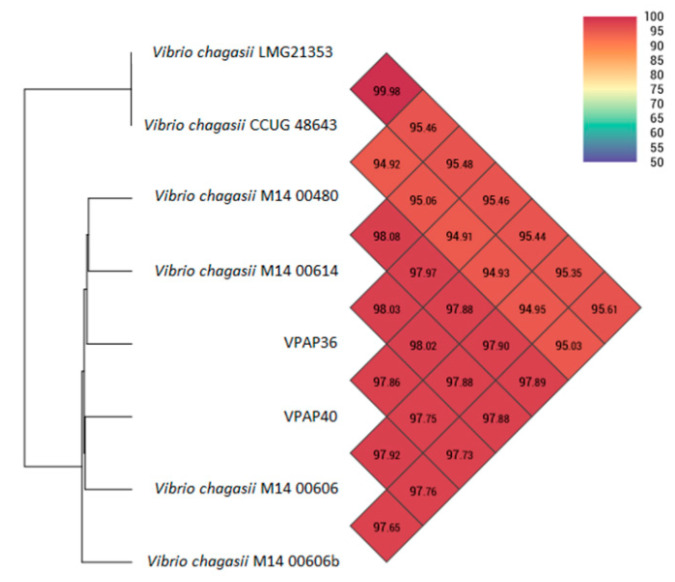
Comparative results of the Average Nucleotide Identity using OrthoANI values of six *Vibrio chagasii* genomes, calculated from the OAT software.

**Figure 4 pathogens-12-00183-f004:**
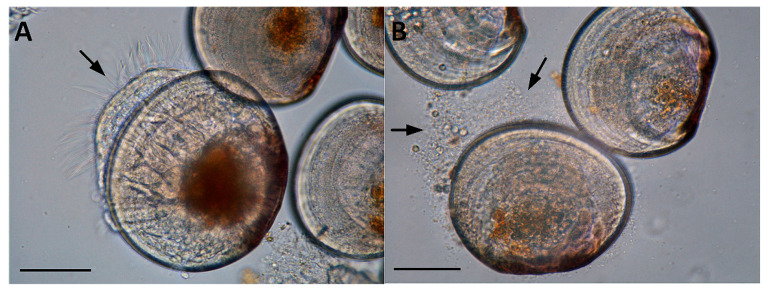
Main symptoms of pathogenic activity of *Vibrio chagasii* VPAP36 on experimentally infected *Argopecten purpuratus* larvae after 24 h exposure. (**A**) distended velum, and (**B**) aggregates of bacteria along with detached larval cells on the margins of the larvae. Scale bars: 50 μm.

**Figure 5 pathogens-12-00183-f005:**
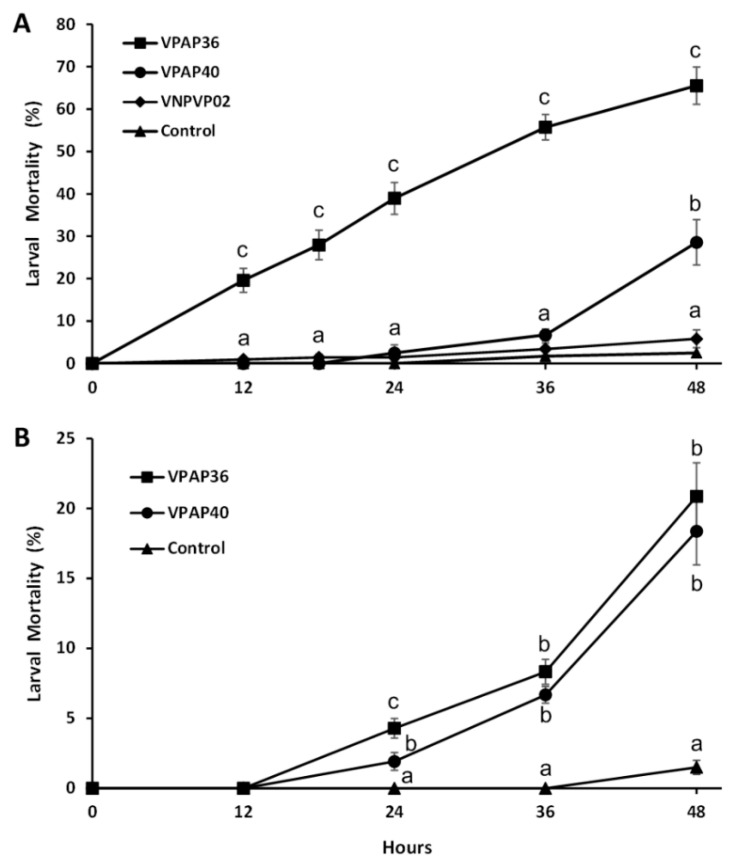
Mortality of 11-day-old scallop larvae unchallenged (control) and challenged with bacteria (**A**) and extracellular products (**B**), of *Vibrio chagasii* strains VPAP36 and VPAP40, and the non-pathogenic *Vibrio* strain VNPAP02, over a 48 h period. Letters indicate significant differences between treatments (mean ± SD, ANOVA *p* < 0.05).

**Table 1 pathogens-12-00183-t001:** Enzymatic properties of the *Vibrio chagasii* VPAP36 and VPAP40 strains using the APIZYM system (BioMerieux).

Enzyme	Activity	
	VPAP36	VPAP40
Control	Negative	Negative
Alkaline phosphatase	Positive	Positive
Esterase (C4)	Negative	Negative
Esterase lipase (C8)	Negative	Negative
Lipase (C14)	Negative	Negative
Leucine arylamidase	Positive	Positive
Valine arylamidase	Positive	Positive
Cystine arylamidase	Negative	Negative
Trypsin	Positive	Positive
α-Chymotrypsin	Negative	Negative
Acid phosphatase	Positive	Positive
Naphthol-AS-BI-Phosphohydrolase	Positive	Positive
α-Galactosidase	Negative	Negative
β-Galactosidase	Negative	Negative
β-Glucoronidase	Negative	Positive
α-Glucosidase	Negative	Negative
β-Glucosidase	Negative	Negative
*N*-Acetyl-β-glucosaminidase	Negative	Negative
α-Mannosidase	Negative	Negative
α-Fucosidase	Negative	Negative

## Data Availability

The whole genome sequences of the VPAP36 and VPAP40 strains have been deposited at DDBJ/ENA/GenBank under the accession numbers JAOZWY000000000 and JAPDFM000000000, respectively.

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
