# Peer review of "First Report, Characterization and Pathogenicity of Vibrio chagasii Isolated from Diseased Reared Larvae of Chilean Scallop, Argopecten purpuratus (Lamarck, 1819)"

_pathogens, 2023, doi:10.3390/pathogens12020183_

Round 1
Reviewer 1 Report
Dear Authors,
Please find my suggestions in the manuscript file.

Author Response
We really appreciate the comments and suggestions of the reviewer, which contributed to improving the article.All recommendations made by the reviewer and included in the submitted pdf file were considered in this corrected version of the manuscript.
Reviewer 2 Report
This paper describes the isolation and characterisation of 2 Vibrio chagasii strains that are pathogenic to Chilean scallop. The work is novel and interesting, conclusions are supported by data and the paper is well-written. Therefore, I only have comments of relatively minor importance to further improve the manuscript.
lines 143-158: this paragraph describes experiments to determine virulence factor production. However, the results of these experiments are not included in the paper. Please either delete this paragraph from the Materials and Methods section or add the results to the Results section.
line 212-213 “Larvae inoculated with the non-pathogenic strain Vibrio VNPVP02 were used as controls.”: it is not completely clear whether the controls were inoculated with live cells of the non-pathogenic strain or with extracellular products of the non-pathogenic strain. The latter would be the most logical control in order to compare the effect of extracellular products of the isolates.
lines 260-261 “strains were compared with several whole genome sequences”: this is misleading as the phylogenetic tree presented in Figure 2 is not based on whole genome sequences, but on sequences of 100 genes (as mentioned in the figure legend). Please rephrase statement.
lines 268-269 “VPAP35 and VPAP37”: this probably should be “VPAP36 and VPAP40”
line 298 “Cholera toxin”: did the genomes of the V. chagasii isolates indeed contain the Cholera toxin? Has this been reported before in non-cholera vibrios? Please add some comment on this to the Discussion section.
lines 312-314 “Main symptoms of pathogenic activity... Swarms of bacteria on the margins of the larvae (bacterial swarming)”: The terms “swarms” and “bacterial swarming” are not well-chosen because swarming refers to a type of motility in bacteria; “bacterial aggregates” would be more appropriate. In fact, the picture (Figure 4B) shows bacterial aggregates surrounding the larvae; this is not necessarily a pathogenicity symptom. It just shows the presence of bacteria in the scallop cultures.
line 334 “they exhibited identical symptoms to those observed during bacterial challenges.”: What were these symptoms? Please specify.
line 365 “V. kanolae”: this should be “V. kanaloae”
line 365 “stains”: this should be “strains”
lines 395-396 “Symptoms produced by infection with V. chagasii strains have also been observed by infections in larvae with strains of V. bivalvicida and V. europaeus species”: Please specify which symptoms.
line 407 “significantly”: this term refers to statistical analyses, which have not been performed to compare the mortalities caused by the isolates to those caused by other vibrios. Please use another term.
lines 409-410 “ECPs produced by the V. chagasii strains were less toxic to larvae than whole bacterial cells”: Please add some discussion about what could explain this difference.
lines 411-412 “suggesting that various virulence factors are required to produce a high degree of larval mortality”: How does the difference in toxicity between ECPs and live cells suggest that various virulence factors are at play? The statement is very general; please discuss this in more depth.
lines 427-428 “a position of coexistence with these pathological processes is assumed”: this is not clear; please rephrase.
line 432 “bioassays”: please specify what bioassays
lines 436-451: this section is not really relevant to this study (it deals with health management, which was not studied in the current manuscript) and thus can be deleted
lines461-462 “Our results demonstrated that V. chagasii are highly (VPAP36 strain) and moderately (VPAP40 strain) pathogenic to scallop larvae”: Please change “pathogenic” into “virulent”. I would rather describe the virulence of VPAP36 as “moderate” (around 60% mortality; there are other vibrios that cause higher mortality) and the virulence of VPAP40 as “low” (around 30%)
Author Response
Please, see attached file.

Round 2
Reviewer 1 Report
Dear Authors,
There are many responses directed by you to the questions by Reviewers. After revision, the manuscript has a good and logical flow.